# Effect of Ducted Multi-Propeller Configuration on Aerodynamic Performance in Quadrotor Drone

**Yi Li [1], Koichi Yonezawa [2] and Hao Liu [1],***

[1]  Graduate School of Engineering, Chiba University, 1-33 Yayoi-cho, Chiba 263-8522, Japan;
     liyi70109@gmail.com
[2]  Civil Engineering Research Laboratory, Central Research Institute of Electric Power Industry, 1646 Abiko,
     Abiko-shi 270-1194, Japan; koichi-y@criepi.denken.or.jp
*   Correspondence: hliu@faculty.chiba-u.jp

**Abstract:** Motivated by a bioinspired optimal aerodynamic design of a multi-propeller configuration, here we propose a ducted multi-propeller design to explore the improvement of lift force production and FM efficiency in quadrotor drones through optimizing the ducted multi-propeller configuration. We first conducted a CFD-based study to explore a high-performance duct morphology in a ducted single-propeller model in terms of aerodynamic performance and duct volume. The effect of a ducted multi-propeller configuration on aerodynamic performance is then investigated in terms of the tip distance and the height difference of propellers under a hovering state. Our results indicate that the tip distance-induced interactions have a noticeable effect in impairing the lift force production and FM efficiency but are limited to small tip distances, whereas the height difference-induced interactions have an impact on enhancing the aerodynamic performance over a certain range. An optimal ducted multi-propeller configuration with a minimal tip distance and an appropriate height difference was further examined through a combination of CFD simulations and a surrogate model in a broad-parameter space, which enables a significant improvement in both lift force production and FM efficiency for the multirotor, and thus provides a potential optimal design for ducted multirotor UAVs.

**Keywords:** ducted multi-propeller configuration; aerodynamic interaction; aerodynamic performance; CFD-based simulation; surrogate model





## 1. Introduction

The quadrotor drone, a type of unmanned aerial vehicle (UAV) or micro air vehicle (MAV) that is capable of vertical take-off and landing (VTOL), has a wide range of applications such as surveillance and reconnaissance in the military field, traffic monitoring and pollution detection in the industrial field, aerial mapping and delivery in the daily life field [1–3], etc., and it increasingly draws much interest in civil applications and academic research due to its advantages of convenient handling characteristics, low cost, and simple maneuverability [4,5]. For the sake of completing various missions outstandingly, it is vital for this aircraft to possess some characteristics such as high efficiency, stability, maneuverability, aerial duration, and so on. Many research studies on multirotor copters have been conducted, associated with lift force and efficiency improvement in the manner of blade optimization design [6], overlapping propellers' design at different heights [7], and multirotor design regarding rotors with a large tip distance [8], tilt [9] or shroud [10–12].

Inspired by insects and birds that achieve high aerodynamic performance and flight control with an optimal combination of a paired-wings configuration [13,14], stroke-plane inclination [15,16], and flapping wings' Euler angles asymmetry [17,18], we recently proposed a biomimetic optimal non-ducted multi-propeller configuration design that is verified to be capable of achieving optimal aerodynamic performance for the quadrotor drone

(DJI phantom 3 advanced in Figure 1a [19]. It points to the maximum non-ducted multi-propeller configuration as depicted in Figure 1b, with a large rotor-to-rotor tip distance, some height difference, and zero tilt angle. This configuration enables optimal aerodynamic interactions among propellers, leading to a marked improvement in lift force production with an increase rate of 9% compared to that of a basic non-ducted multi-propeller configuration (Figure 1a), thus enhancing the FM efficiency.

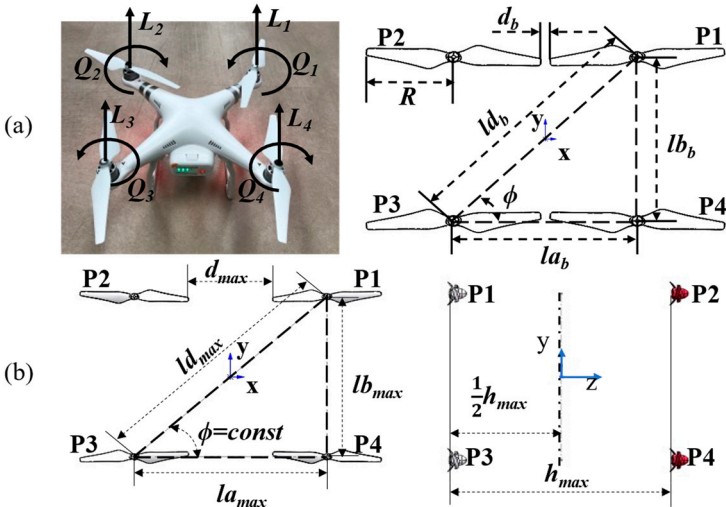

**Figure 1.** Illustration of a quadrotor copter, propeller configuration, and morphological parameters. (**a**) Quadrotor copter of DJI phantom 3 advanced with definitions of lift force ($L$), torques ($Q$), and a basic propeller configuration with $R$ = 0.12 m, $la_b \approx$ 0.252 m, $lb_b \approx$ 0.243 m, $ld_b$ = 0.35 m, $d_b \approx$ 0.012 m, $\varnothing \approx 44°$ at height difference, $h_b = 0$. (**b**) The maximum (optimal) propeller configuration with $la_{max} \approx$ 0.425 m, $lb_{max} \approx$ 0.410 m, $ld_{max}$ = 0.59 m, $d_{max} \approx$ 0.185 m, $\varnothing \approx 44°$ at height difference, $h_{max}$ = 0.24 m.

To further improve the aerodynamic performance of the quadrotor drones, we chose the approach of employing the ducted-propeller design [20] while adopting the multi-propeller configuration [21] to optimize the lift force production and FM efficiency. The aerodynamic optimization of the ducted-propeller configuration thus consists of minimizing the propeller–duct interactions, the reduction of the duct's weight [22], the duct geometry, the tip clearance between propeller tip and the duct inwall, and so on. With respect to the ducted-propeller application in UAVs and MAVs, duct (shroud) designs with various blades, shroud dimensions, and height difference have been proposed and developed, for instance, to evaluate the hovering performance and edgewise flow experimentally [10–12] and computationally [23,24]. Chua, et al. [25] computationally and experimentally investigated the effect of shrouded rotors on energy cost/efficiency and thrust power by altering the leading-edge lip radius (LLR), the diffuser length (DL), and the diffuser angle (DA). Moaad, et al. [26] reported that the improvement of drone maneuverability can be achieved by optimizing the ducted fan. Penkov, et al. [27] examined the interactive impact on the lift force production on a mini UAV with the propellers shrouded in various shroud diameters and heights computationally and experimentally. Shukla, et al. [28] conducted aeroacoustic measurements with a rotor with a removable protective duct over a range of hover conditions by means of stereo particle image velocimetry (SPIV). Recently, Yonezawa, et al. [29,30] investigated the aerodynamic characteristics of ducted single-propellers in various duct contours and ducted multi-propellers in a quadrotor drone under hovering condition with and without crosswind, experimentally and numerically. Until now, most studies have been focused on the aerodynamic optimization of ducts with some fixed propeller-configuration, aiming to improve lift force production and/or maneuverability of multirotor copters. It still remains poorly understood how the different configurations of ducted multi-propeller affect the aerodynamic performance of the multirotor copter associated with lift force production and FM efficiency in terms of tip distance

and height difference adjustment, as well as which is the optimal ducted multi-propeller configuration with these two parameters.

In this study, we perform an integrated simulation-based study of CFD simulations and a surrogate model to investigate the effect of the ducted multi-propeller configuration on aerodynamic performance and to explore the optimal ducted multi-propeller configuration of a quadrotor drone. We first explore a high-performance and compact duct design in terms of aerodynamic performance and duct volume based on CFD results of ducted single-propeller models. The duct design obtained is then adopted to a quadrotor drone to examine the effect of the ducted multi-propeller configuration on aerodynamic performance associated with tip distance and height difference among various ducted propellers. Furthermore, an extensive analysis of the optimal ducted multi-propeller configuration is conducted through the combination of CFD simulations and a surrogate model to search for the optimized design in a broad-parameter space of the tip distance and height difference, which is verified to improve both lift force production and FM efficiency.

## 2. Materials and Methods

### 2.1. Aerodynamic Theory of a Ducted Propeller

Estimation of the aerodynamic lift force on an isolated propeller in a ducted-propeller model is derived from the momentum theory, the blade element theory, and the aerodynamic principle of a ducted propeller [9,19,31,32] (Figure 2). The duct is composed of a rounded leading edge and a straight or tapered trailing edge formed as the inlet and diffuser section, respectively. The rotor operation generates a suction pressure gradient on the duct inlet surface, thus resulting in additional lift, which contributes to the total lift force and hence enhances the FM efficiency [10–12].

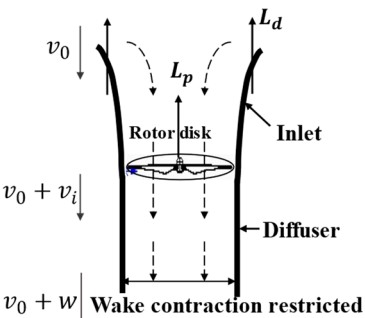

**Figure 2.** Schematic diagram of the ducted-propeller aerodynamic principle. Rotor disk and downwash in hovering state: Lift force generated by propeller ($L_p = 2\rho A v_i^2$), $v_0 = 0$, induced velocity ($v_i$), and far wake velocity ($w = 2v_i$), additional lift force generated by duct ($L_d$).

### 2.2. Geometric Model of Ducted Propeller

#### 2.2.1. Ducted Single-Propeller Geometry

The duct is generally composed of a straight diffuser section and an elliptic or pseudo-elliptic inlet (Figure 3), which is verified to enable a significant improvement in lift/thrust force production and power reduction particularly at low rotational speeds and/or with high disk-loading [10–12]. Therefore, in this study, the duct with an ellipse inlet is employed and the aerodynamic performance of ducted-propeller units is discussed extensively. To determine a high-performance duct design in the ducted single-propeller model in terms of the duct's cross-section and tip clearance, six parameters in toto are utilized, including the tip clearance ($d_e$) expressing the gap between propeller tip and duct inwall; the propeller height ($h_p$), i.e., the height difference between the center of the inlet ellipse (point D) and the center of the propeller bottom (point P); the diffuser angle ($\alpha$) denoting the inclination angle of the diffuser; the diffuser length ($l_e$); the height of ellipse inlet ($h_e$); and the radius of the ellipse inlet ($r_e$), with an original value of $d_e = 0.001$ m, $h_p = 0$, $\alpha = 0$, $l_e = 0.06$ m (0.5R), $h_e = 0.06$ m (0.5R), and $r_e = 0.02$ m (0.167R) (Figure 3). The thickness of the duct model is

fixed to be 0.0015 m. The 3D single-propeller model is based on DJI phantom 3 advanced (Figure 1a) as used in our previous studies [19,33]. A systematic CFD-based analysis was conducted to examine the high-performance duct model in terms of aerodynamic performance and duct volume.

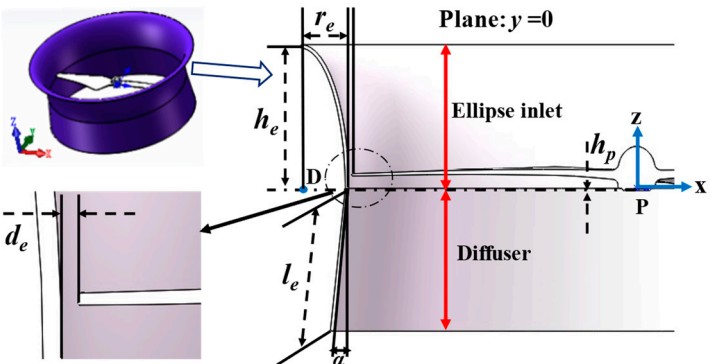

**Figure 3.** Morphological parameters of the ducted single propeller model. Tip clearance ($d_e$), propeller height ($h_p$ : Being positive when point D is beyond point P and vice versa), diffuser angle ($\alpha$: Being positive with inclination outward but negative with inclination inward), diffuser length ($l_e$), height of ellipse inlet ($h_e$), and radius of ellipse inlet ($r_e$).

### 2.2.2. Ducted Multi-Propeller Geometry

The ducted multi-propeller model is generated based on the duct model obtained in Section 2.2.1. Figure 1 shows the geometric models of a basic non-ducted multi-propeller configuration (Figure 1a) and a maximum non-ducted multi-propeller configuration (Figure 1b) obtained in our previous research [19], which is the optimal configuration with a maximum rotor-to-rotor tip distance, verified to be capable of improving both lift force production and FM efficiency in the quadrotor drone of DJI phantom 3 advanced. Given the high-performance duct model as an initial input to the ducted multi-propeller configuration with the maximum rotor-to-rotor tip distance, the ducted maximum multi-propeller configuration (Figure 4a) is formed, and the ducted minimum multi-propeller configuration (Figure 4b) is generated through altering the rotor-to-rotor tip distance and height difference. Based on the analysis of lift force production and FM efficiency in a broad parameter space associated with the tip distance and height difference (see Sections 3.2 and 3.3), the effect and optimization of a ducted multi-propeller configuration on aerodynamic performance is investigated based on a combination of CFD simulations and a surrogate model.

### 2.3. CFD Modeling

CFD-based simulations were conducted with the commercial software ANSYS CFX 14.5 (ANSYS Inc) under the conditions of a rotational speed of 5400 rpm for all propeller models with/without ducts, which is identical to that utilized in our previous studies [19,33]. The Reynolds number (*Re*) of a single propeller is calculated to be $7.4 \times 10^4$ [19], and the RANS modelling of turbulent flow with the SST turbulence model was adopted with a 'high-resolution mode' for all the simulations of ducted single-propeller and ducted multi-propeller models [19,33,34]. Following our previous studies [19,33,34], we generated the inflation layer meshes with seven layers surrounding the propeller surfaces to ensure high resolution of the boundary layer adjacent to walls while being clustered at the wingtip, leading edge, and trailing edge. Approximately 26 million and 48 million meshes (see in Sections 3.1 and 3.2) were successively generated for the ducted single-propeller and ducted multi-propeller, respectively. Furthermore, boundary conditions and grid systems are given in Figure 5. A "Frozen Rotor (FR)" approach was utilized at the interface between inner rotating and outer stationary regions of both ducted single-propeller and multi-propeller models to give the rotor an appointed constant ro-

tating speed for the sake of 'freezing' the relative movements between the two frames, which thus ensures convergence to a stable state. Besides, a 'General Connection with No Frame Change/Mixing' model was employed at the interface between intermediate stationary regions and coarse stationary regions of ducted multi-propeller models for connection. In addition, the wall boundary was used at the duct surfaces, and the open (free-stream boundary) condition with 0 Pa pressure was adopted at the outside boundary of the spherical surface.

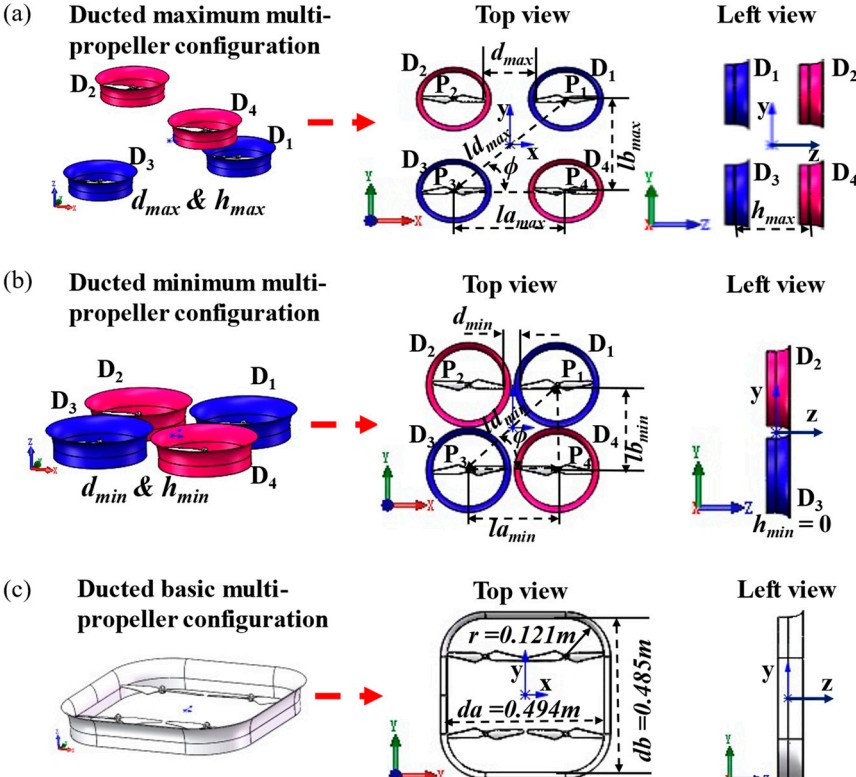

**Figure 4.** Definitions of various configurations of the ducted multi-propeller model. (**a**) Ducted maximum multi-propeller configuration with $d_{max} \approx 0.185$ m, $h_{max} = 0.24$ m, $la_{max} \approx 0.425$ m, $lb_{max} \approx 0.410$ m, $ld_{max} = 0.59$ m at an inclination angle of $\varnothing \approx 44°$. (**b**) Ducted minimum multi-propeller configuration with $d_{min} \approx 0.055$ m, $h_{min} = 0$, $la_{min} \approx 0.295$ m, $lb_{min} \approx 0.285$ m, $ld_{min} = 0.41$ m at an inclination angle of $\varnothing \approx 44°$. (**c**) Ducted basic multi-propeller configuration with $da = 0.494$ m, $db = 0.485$ m, and $r = 0.121$ m.

FM efficiency [19,33] is adopted to evaluate the aerodynamic performance of the ducted single-propeller model and defined as:

$$FM_{D-SP} = \frac{P_{RF,D-SP}}{P_{CFD,D-SP}} \tag{1}$$

where $P_{RF,D-SP}$ denotes the minimum power derived from the Rankin–Froude momentum theory for generating lift force based on numerical results. $P_{CFD,D-SP}$ is calculated from a product of the torque around rotational axis, $Q$, and the rotational angular velocity, $\omega$, which are formulated as:

$$\begin{cases} P_{RF,D-SP} = L_{D-SP}\sqrt{\frac{L_{D-SP}}{2\rho A_{SP}}} \\ P_{CFD,D-SP} = Q \cdot \omega \end{cases} \tag{2}$$

where $L_{D-SP}$ expresses the lift force of the ducted single propeller ($L_{D-SP} = L_P + L_D$, $L_P$: Lift force on propeller; $L_D$: Lift force on duct). $A_{SP}$ is the actuator disk's area defined by the propeller's radius, $R$, and $\rho$ is the air density.

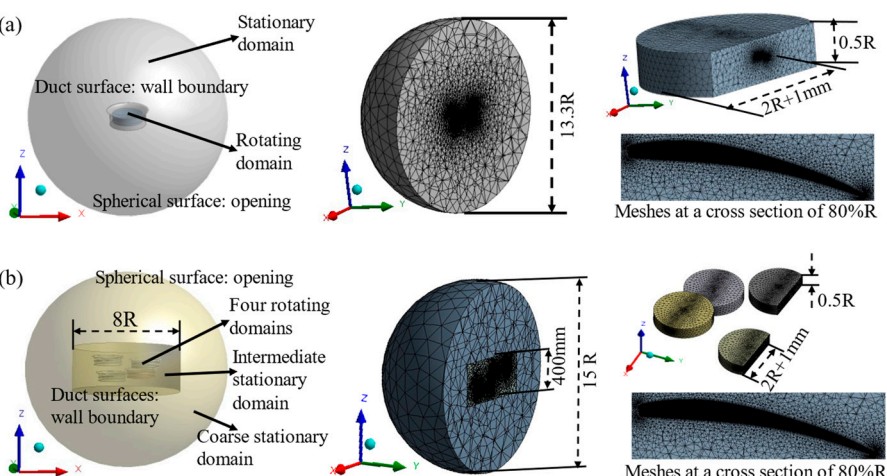

**Figure 5.** Mesh systems and boundary conditions for CFD simulations. (**a**) Ducted single-propeller model. (**b**) Ducted multi-propeller model.

For the ducted multi-propeller models with various configurations of tip distance and height difference, Equations (1–2) can be further used in the evaluation of aerodynamic performance with some refinements, where the FM efficiency of a ducted multi-propeller model is defined as $FM_{D-MP}$, with the $L_{D-SP}$ substituted for the ducted-multi-propeller-induced lift force, $L_{D-MP}$ ($L_{D-MP} = L_{MP} + L_{MD}$, $L_{MP} = L_{P1} + L_{P2} + L_{P3} + L_{P4}$, $L_{MD} = L_{D1} + L_{D2} + L_{D3} + L_{D4}$), the $A_{SP}$ by $A_{MP}$ ($A_{MP} = 4A_{SP}$), and the $P_{CFD,D-SP}$ by $P_{CFD, D-MP}$ ($P_{CFD}$ of ducted multi-propeller: $P_{CFD,D-MP} = Q_1 \cdot \omega_1 + Q_2 \cdot \omega_2 + Q_3 \cdot \omega_3 + Q_4 \cdot \omega_4$), respectively. Comparatively, regarding the ducted basic multi-propeller configuration model with one duct surrounding the outside, the $L_{D-SP}$ will be replaced by the lift force, $L_{D-BMP}$ (lift force of ducted basic multi-propeller: $L_{D-BMP} = L_{BMP} + L_{D-B} = L_{P1} + L_{P2} + L_{P3} + L_{P4} + L_{D-B}$, $L_{D-B}$: Lift force on duct surrounding outside).

### 2.4. Optimization of Aerodynamic Performance in Ducted Multi-Propeller Configuration

CFD-based simulations of 22 cases (see Section 3.2) with various ducted multi-propeller configurations associated with different tip distances and height differences were conducted to examine the effect of ducted multi-propeller configurations on their aerodynamic performance. Furthermore, a surrogate model combined with a set of CFD-based cases (35 cases in total, see Section 3.3) was employed to explore the optimal ducted multi-propeller configuration in terms of $L_{D-MP}$ and $FM_{D-MP}$ in a broad parametric space of tip distance and height difference.

The surrogate model with an alternative interpolation method of the Radial Basis Functions (RBFs) model is a versatile while fast optimization method [35]. It is implemented here in three steps: (1) Specification of a design space based on CFD-based numerical experiments comprising 35 discrete points associated with two parameters (tip distance and height difference); (2) CFD simulations at the design points; and (3) construction of a surrogate model based on the CFD simulations to achieve a continuous output over the entire design space [19,36]. As a result, a continuous map as a continuous spatial surface of $L_{D-MP}$ or $FM_{D-MP}$ will be yielded in the parametric space.

For the RBFs approach, $L_{D-MP}$ or $FM_{D-MP}$ is approximated as an unknown function of $f(x)$ at an untried point, $x$, with a linear combination of radial basis functions, defined as:

$$f(x) = \sum_{i=1}^{n} w_i \varphi(r) \qquad (3)$$

where $w_i$ is the $i$-th weight coefficient, and $\varphi(r) = \varphi(\|x^i - x\|)$ is the basic function determined by the Euclidean distance between the prescribed observed point $x^i$ and the untried point $x$ [19,37,38]. To determine the weight coefficient $w_i$, a set of interpolation points of $x^j$ that have known results from CFD simulations are introduced to substitute the untried points of $x$, where all the interpolation points should satisfy:

$$f\left(x^j\right) = \sum_{i=1}^{n} w_i \varphi(r) = \sum_{i=1}^{n} w_i \varphi\left(\|x^i - x^j\|\right) = y^j, \; j = 1, 2, \ldots, n, \tag{4}$$

where $x^j$ denotes the interpolation point, and $y^j$ is the result at the corresponding interpolation point, respectively. Thus, with the known observed points and interpolation points, the weight coefficient $w_i$ can be determined subsequently. In order to determine the location of interpolation points in the design space, the Uniform Design (UD) method [36] of the Design of Experiments (DoE) is employed in advance for the determination, which can maximize the amount of information from a limited number of CFD-based results at the sampled discrete points, i.e., 35 CFD-based $L_{D-MP}$ or $FM_{D-MP}$. Consequently, nine interpolation points ($n = 9$) are determined and utilized, which is verified to be effective enough to provide a reasonable approximation based on our previous study [19]. Besides, the optimal observed points are determined in the vicinity of the interpolation points correspondingly, in a manner of random selection.

The Inverse Multiquadric (IMQ) function is adopted through trial-and-error from multiple basic function options in the RBFs model [19,37,38] and utilized in Equation (3), which is capable of providing reasonable results in approximating the lift force at all points, and is defined as:

$$\varphi(r) = 1/\sqrt{r^2 + c^2}, \; 0 < c < 4 \tag{5}$$

With a certain coefficient of $c$ in the basic function of $\varphi(r)$, the weight coefficient of $w_i$ is determined by solving Equation (3) based on the interpolation points and observed points. The consistency between CFD- and surrogate model-based results is verified through a comparison of three variables of 35 test points between CFD and surrogate model based results. The comparison is conducted by utilizing the average relative error ($\bar{e}$), the R-squared ($R^2$), and the root mean squared error ($\sigma_e$) with the convergence criteria of $\bar{e} < 0.0025$, $R^2 > 0.945$, and $\sigma_e < 0.0025$ [19,37,38], where $\bar{e}$ is defined as:

$$\bar{e} = \frac{1}{n_t} \sum_{i=1}^{n_t} e^i = \frac{1}{n_t} \sum_{i=1}^{n_t} \|\frac{\hat{y}^i - y^i}{y^i}\| \tag{6}$$

where $n_t$ denotes the number of test points ($n_t = 35$), $y^i$ is the CFD-based result, and $\hat{y}^i$ is the predicted result at the $i$-th test point from the surrogate model, respectively. The $R^2$ is defined as:

$$R^2 = 1 - \frac{\sum_{i=1}^{n_t} \left(y^i - \hat{y}^i\right)^2}{\sum_{i=1}^{n_t} \left(y^i - \bar{y}\right)^2} \tag{7}$$

where $\bar{y}$ is the average of CFD-based results. Furthermore, $\sigma_e$ is defined as:

$$\sigma_e = \sqrt{\frac{1}{n_t} \sum_{i=1}^{n_t} \left(e^i\right)^2} \tag{8}$$

An inner iteration is used to inspect the optimal variables of the coefficient, $c$, varying over a predetermined parametric range (Equation (5)), and a flowchart associated with the optimization based on CFD simulations and the surrogate model is shown in Figure 6.

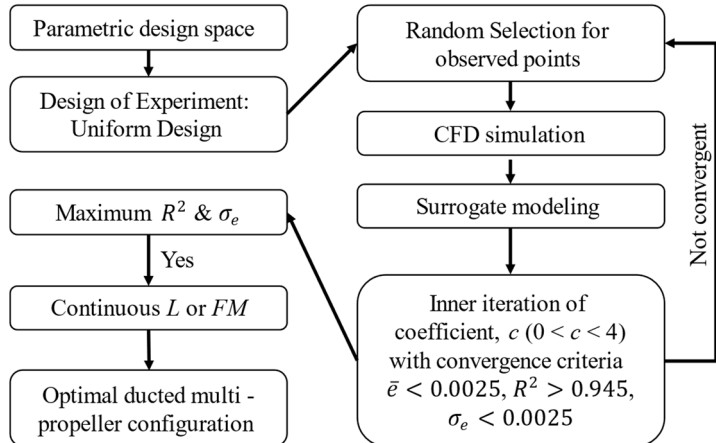

**Figure 6.** Flow chart of CFD- and surrogate model-based optimization analysis.

## 3. Results and Discussion

### 3.1. High-Performance Duct Design in Ducted Single-Propeller Model

3.1.1. Verification and Validation

Verification was carried out via self-consistency on mesh independency. The minimum grid spacing adjacent to the wall (propeller) surface, $\delta_m$, is controlled by $\delta_m \approx 0.1l/\sqrt{Re}$, where $Re = 7.4 \times 10^4$, $l = 0.0162$ m (chord length at 75%$R$) [19,34], resulting in the minimum grid spacing of 0.015 mm. We composed a baseline case of the ducted single-propeller model (D-SP-1), employed the basic duct (Table 1) with approximately 26 million meshes [19], and two other cases of 39 million meshes (D-SP-2) and 19 million meshes (D-SP-3), which were compared in terms of computed lift forces and FMs, as shown in Figure 7. A marginal difference in both lift forces and FMs is found among the three cases. Thus, we employed the mesh setting of D-SP-1 in all the other CFD simulations of ducted single-propeller models with the consideration of computer time and numerical accuracy. Besides, the criterion of numerical convergence was set to be that either the maximum residual of pressure is less than $5 \times 10^{-5}$ or the maximum iteration steps is more than 3000.

**Table 1.** Morphology parameters in the basic duct model and the high-performance duct model.

|  | $d_e$ (m) | $h_p$ (m) | $\alpha$ (°C) | $h_e/R$ | $l_e/R$ | $r_e/R$ |
|---|---|---|---|---|---|---|
| Basic duct | 0.001 | 0 | 0 | 0.5 | 0.5 | 0.167 |
| High-performance duct | 0.001 | 0 | 0 | 0.375 | 0.25 | 0.167 |

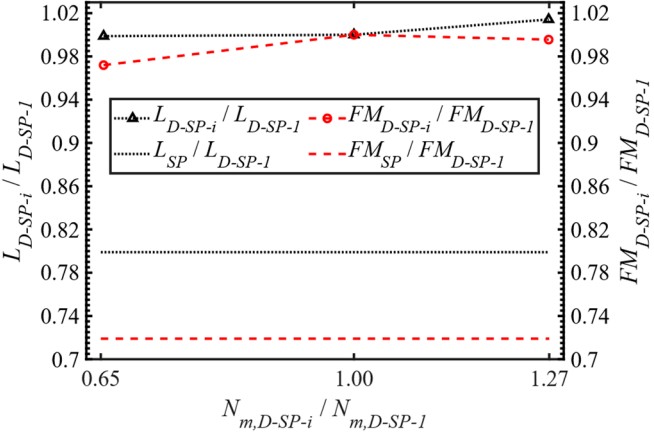

**Figure 7.** Comparison of lift forces and FMs among three grid systems in the ducted single-propeller model ($N_m$: Mesh number; $L_{SP}$ = 3.11N, $FM_{SP}$ = 0.64 [19,33]).

The CFD simulations were further validated via comparison of the lift force generated by the propeller with EXP data [27] under the same conditions, in terms of different diameters of the cylindrical shroud surrounding the propeller, from 270 mm to 310 mm with an interval of 10 mm, and a fixed shroud height of 60 mm (with the propeller having a radius of 127 mm and a rotational velocity of 5000 rpm). As shown in Figure 8, the CFD results are consistent with EXP results, sharing a similar trend that the propeller-induced lift force increases linearly with an increasing gap, while being lower than that of the non-ducted propeller accompanying a notable difference, which is caused by the different propeller morphology in the models of CFDs and EXPs. Moreover, it is worth noting that a propeller–duct aerodynamic interaction exists, which may significantly affect the aerodynamic performance of the ducted-propeller associated with lift force production and FM efficiency.

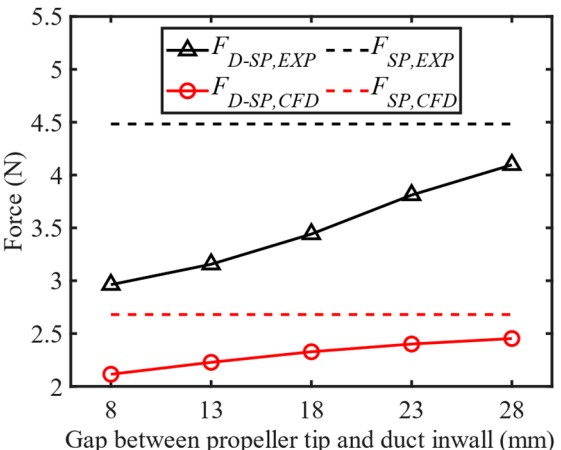

**Figure 8.** Comparison of lift forces between CFD and EXP [27] in the ducted single-propeller model. $F_{D-SP,EXP}$ and $F_{D-SP,CFD}$, lift force generated by propeller in the ducted single-propeller model (EXP and CFD); $F_{SP,EXP}$ and $F_{SP,CFD}$, lift force generated by propeller in the non-ducted single-propeller model (EXP and CFD).

### 3.1.2. High-Performance Duct Design

A high-performance duct design for the single-propeller model was first explored in terms of lift force ($L_{D-SP}$) and FM efficiency ($FM_{D-SP}$) based on a variety of CFD simulations through adjusting one parameter while keeping others fixed, regarding the six parameters shown in Figure 3. With consideration of the limitation about duct weight, the duct volume that is proportional to duct weight was taken as an additional parameter and should be reduced as much as possible, simultaneously.

Figure 9 shows the correlations between aerodynamic performances ($L_{D-SP}$ and $FM_{D-SP}$) and the variation of tip clearance ($d_e$), height difference ($h_p$), and diffuser angle ($\alpha$). While lift force and FM efficiency show significant dependency upon tip clearance (Figure 9a), some maxima in $L_{D-SP, d_e}$ and $FM_{D-SP, d_e}$ are found at $d_e = 0.001$ m with other parameters fixed at $h_p = 0$, $\alpha = 0$, $h_e = 0.06$ m, $l_e = 0.06$ m, $r_e = 0.02$ m, resulting in a marked increase rate of 25.1% in lift force and an increase rate of 39.1% in FM efficiency compared to those of the non-ducted single-propeller model ($L_{SP}$ and $FM_{SP}$). The height difference dependency of $L_{D-SP, h_p}$ or $FM_{D-SP, h_p}$ (Figure 9b) is moderate, with their maxima around $h_p = 0$ accompanying other parameters fixed at $d_e = 0.001$ m, $\alpha = 0$, $h_e = 0.06$ m, $l_e = 0.06$ m, and $r_e = 0.02$ m. The diffuser angle seems to affect $L_{D-SP, \alpha}$ and $FM_{D-SP, \alpha}$ (Figure 9c) significantly, leading to the maxima at $\alpha = 0$ with other parameters fixed at $d_e = 0.001$ m, $h_p = 0$, $h_e = 0.06$ m, $l_e = 0.06$ m, and $r_e = 0.02$ m. It is worth noting that the variation of these three parameters hardly changes the duct volume, thus the duct volume is not taken into account for these three parameters in the current case.

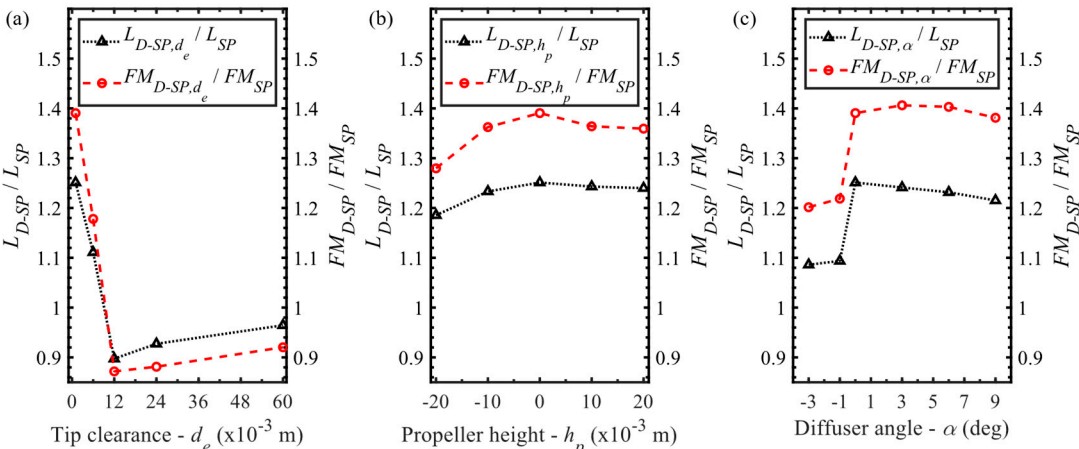

**Figure 9.** Lift force and FM efficiency vs. (**a**) tip clearance ($d_e$) ($L_{D-SP, d_e}$ and $FM_{D-SP, d_e}$), (**b**) height difference ($h_p$) ($L_{D-SP, h_p}$ and $FM_{D-SP, h_p}$), and (**c**) diffuser angle ($\alpha$) ($L_{D-SP, \alpha}$ and $FM_{D-SP, \alpha}$) in the ducted single-propeller model. $L_{SP}$ and $FM_{SP}$ respectively denote the lift force and FM efficiency in the non-ducted single-propeller model.

Figure 10a shows the variations of $L_{D-SP}$ and $FM_{D-SP}$ in various combinations of the dimensionless diffuser length ($l_e/R$) and dimensionless height of ellipse inlet ($h_e/R$) while keeping other parameters fixed at $d_e = 0.001$ m, $h_p = 0$, $\alpha = 0$, and $r_e = 0.02$ m. Since the duct volume varies with different combinations of $l_e/R$ and $h_e/R$, we further draw a comparison between the increase rates associated with lift force ($Ra_L$: $L_{case-i}/L_{case-1}$) and duct volume ($Ra_V$: $V_{case-i}/V_{case-1}$) in Figure 10b among the four cases of different $l_e/R$ and $h_e/R$ that have large values in $L_{D-SP}$ and $FM_{D-SP}$ as shown in Figure 10a. Obviously, the duct model with $h_e/R = 0.375$ and $l_e/R = 0.25$ ($h_e = 0.045$ m, $l_e = 0.03$ m) is a high-performance duct design capable of achieving the best aerodynamic performance with a minimal duct weight based on the difference between the increase rates of lift force and duct volume ($D_{Ra}$: $Ra_L - Ra_V$) (Figure 10b). Moreover, the ducted single-propeller model with an ellipse inlet height of $h_e = 0.045$ m and a diffuser length of $l_e = 0.03$ m shows a marked improvement on aerodynamic performance with an increase rate of 24.5% in $L_{D-SP}$ and 38.1% in $FM_{D-SP}$ compared to $L_{SP}$ and $FM_{SP}$. With respect to the $r_e$ effect on aerodynamic performance ($L_{D-SP,r_e}$ and $FM_{D-SP,r_e}$) (Figure 11a), while a monotonic increase is observed with increasing $r_e$ with other parameters fixed at $d_e = 0.001$ m, $h_p = 0$, $\alpha = 0$, $h_e = 0.045$ m, and $l_e = 0.03$ m, it also results in the increase of duct volume based on the $Ra_{V, r_e}$ ($V_{r_e, case-i}/V_{r_e, case-1}$) (Figure 11b). Thus, with consideration of the factors of both lift force and duct weight, we propose that the duct model with $r_e = 0.02$ m can be a high-performance duct design, which noticeably leads to the peak of $D_{Ra, r_e}$ ($Ra_{L, r_e} - Ra_{V, r_e}$, $Ra_{L,r_e} = L_{r_e, case-i}/L_{r_e,case-1}$).

Our results indicate that the duct can alter the propeller-induced tip vortex in a manner of duct–propeller interaction, resulting in an enhancement of lift force production in a very small value of tip clearance, which can generate an additional lift force because the pressure difference between the inner and outer surfaces of the duct can induce a suction pressure gradient on the inlet surface (Figure 2). However, such propeller–duct aerodynamic interactions will be weakened with increasing tip clearance ($d_e$), substantially approaching the lift force production of a non-ducted single propeller model ($L_{SP}$). Besides, the variations of other parameters can also alter the propeller-induced tip vortices or wake contraction (Figure 2) due to the duct–propeller interaction, and thus affect the aerodynamic lift force production and FM efficiency. These results are supported by the visualized flow fields and pressure distributions (Figure 12) of the non-ducted single propeller model and the high-performance ducted single-propeller model, where the downwash in the ducted-propeller model is weakened compared to that in the non-ducted model (Figure 12a), which results in reducing the propeller-induced lift force, whereas the pressure gradient on the inlet surface augments the lift force production by the duct. Thus, the duct leads

to improving the total lift force production in the ducted-propeller model. Obviously, the aerodynamic interaction between the propeller and duct plays a crucial role in dominating the tip vortex, the downwash, and the wake topology, exhibiting distinguished features at the tip-gap between the propeller and duct. Thus, the duct enables a significant suppression of the tip vortex while forming a highly contracted yet intense downward jet below the propeller, leading to the enhancement of aerodynamic performance.

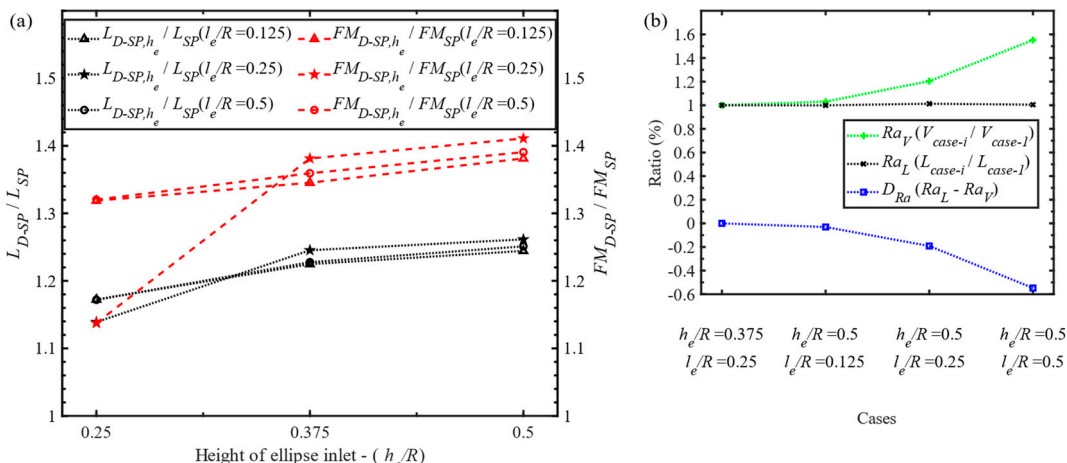

**Figure 10.** (**a**) Lift force and FM efficiency ($L_{D-SP, h_e}$ and $FM_{D-SP, h_e}$) vs. $h_e/R$ with $l_e/R$ fixed in different values in the ducted single-propeller model; (**b**) increase rates of duct volume and lift force vs. $h_e/R$ and $l_e/R$.

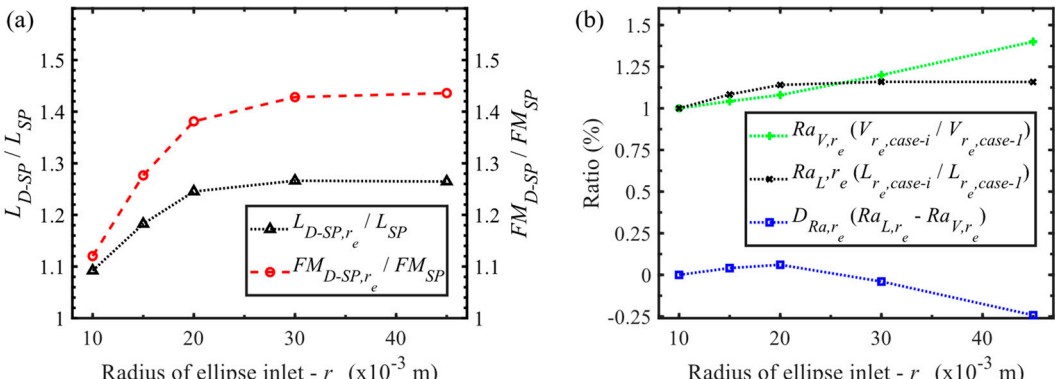

**Figure 11.** (**a**) Lift force and FM efficiency ($L_{D-SP, r_e}$ and $FM_{D-SP, r_e}$) vs. $r_e$ in the ducted single-propeller model; (**b**) increase rates of duct volume and lift force vs. $r_e$.

Thus, the high-performance duct design in the ducted single-propeller model is defined with a combination of $d_e = 0.001$ m, $h_p = 0$, $\alpha = 0$, $h_e = 0.045$ m ($h_e/R = 0.375$), $l_e = 0.03$ m ($l_e/R = 0.25$), and $r_e = 0.02$ m ($r_e/R = 0.167$) (Table 1), which enables a marked improvement in aerodynamic performance with an increase rate of 24.5% in lift force production (3.873N) and an increase rate of 38.1% in FM efficiency (0.884) compared to that in the non-ducted single propeller model. This duct model is subsequently employed in all the ducted multi-propeller models for investigating the configuration effect on aerodynamic performance.

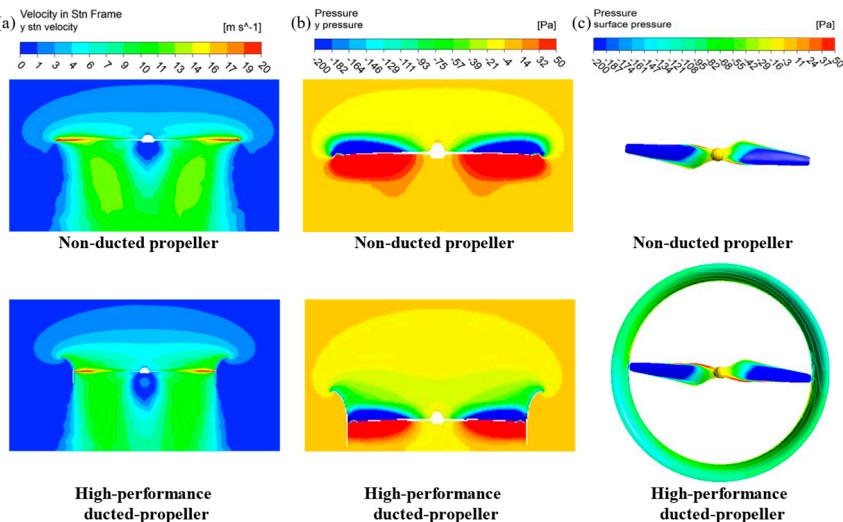

**Figure 12.** Comparison of flow structures and pressure distributions between the non-ducted and high-performance ducted single propeller model. (**a**) Iso-speed contours and (**b**) pressure contours at the cross-section of $y = 0$ m, and (**c**) pressure contours at the suction side.

### 3.2. Effect of Ducted Multi-Propeller Configuration

In our previous work [19], we carried out an extensive study on the effect of a non-ducted multi-propeller configuration on aerodynamic performance in a quadrotor drone, where a combination of tip distance, $d = 0.185$ m ($d/R = 1.54$) and height difference, $h = 0.24$ m ($h/R = 2.0$) identical to the maximum multi-propeller configuration in Figure 1b was found capable of achieving the optimal aerodynamic performance. The optimal configuration achieved the greatest increase rate of 9% in lift force compared with a basic non-ducted multi-propeller configuration under a hovering state. Here, with a series of CFD-based simulations, we intend to examine the effect of a ducted multi-propeller configuration on aerodynamic performance in the quadrotor drone. We employ the high-performance duct model obtained in 3.1 and conduct a systematic parameter study through adjusting the tip distance ($0.46 \leq d/R \leq 1.54$) and height difference ($0 \leq h/R \leq 2.0$) over a broad range between the maximum multi-propeller configuration and the minimum multi-propeller configuration as depicted in Figure 4, where the ducted maximum multi-propeller configuration has a combination of $d = 0.185$ m ($d/R = 1.54$) and $h = 0.24$ m ($h/R = 2.0$) while the minimum one consists of $d = 0.055$ m ($d/R = 0.46$) and $h = 0$ m ($h/R = 0$) that is confirmed to be capable of avoiding the ducted multi-propeller interference. In the end, the CFD-based simulations corresponding to 22 randomly selected combinations as summarized in Table 2 are performed.

**Table 2.** Parameters of $h/R$ and $d/R$ for 22 CFD simulations in various ducted multi-propeller configurations.

| $h/R$ | $d/R$ | | | | | | | |
|---|---|---|---|---|---|---|---|---|
| $h/R = 2.0$ | 1.54 | 1.40 | 1.18 | 1.00 | 0.82 | 0.68 | 0.53 | 0.46 |
| $h/R = 1.5$ | 1.54 | - | - | - | - | - | - | 0.46 |
| $h/R = 1.0$ | 1.54 | - | - | - | - | - | - | 0.46 |
| $h/R = 0.5$ | 1.54 | - | - | - | - | - | - | 0.46 |
| $h/R = 0.0$ | 1.54 | 1.40 | 1.18 | 1.00 | 0.82 | 0.68 | 0.53 | 0.46 |

The modeling validity was first investigated in terms of mesh-dependency associated with the ducted basic multi-propeller configuration (D-BMP), the ducted maximum multi-propeller configuration with $d/R = 1.54$ and $h/R = 2.0$ (D-MMP), and the ducted sub-maximum multi-propeller configuration with $d/R = 1.54$ and $h/R = 0$ (D-SMP). We confirmed that a grid system with 48 million meshes could ensure a good

balance between sufficient numerical accuracy and computation time and thus was used for all the CFD-based ducted multi-propeller simulations. Furthermore, we found that the ducted multi-propeller models in all the multi-propeller configurations show better aerodynamic performance than the non-ducted multi-propeller models with increase rates of lift force and FM efficiency of 7.0% and 9.7% in D-BMP, 15.5% and 24.0% in D-MMP, and 17.7% and 28.0% in D-SMP, respectively. This is also consistent with previous work [29,30].

The effect of ducted multi-propeller configurations was investigated by analyzing the aerodynamic performance associated with lift force ($L_{D-MP}$) and FM efficiency ($FM_{D-MP}$) through adjusting the height difference and tip distance. The height-difference effect was examined as illustrated in Figure 13a through decreasing the dimensionless height difference ($h/R$) with the tip distance fixed. The $L_{D-MP}$ obviously displays some optimal peaks with an increase rate of 1.7% at $h/R = 0.5$ with a fixed tip distance at $d_{max}$, and 0.4% at $h/R = 1.5$ with a fixed tip distance at $d_{min}$, compared to that in the ducted maximum multi-propeller configuration, resulting in the increase rates of 3.9% and 1.7% in $FM_{D-MP}$ at the same points, respectively. This indicates that the $L_{D-MP}$ and $FM_{D-MP}$ can be improved at $h/R$ within a range of 0.5 to 1.5 with the tip distance fixed, particularly when the tip distance is fixed at a larger value. This is probably because the aerodynamic interactions between the lower and upper ducted propellers can enhance the lift force production owing to the increase in induced velocity in the ducted propellers positioned lower when $h/R$ varies from 0.5 to 1.5. On the other hand, the tip-distance effect of ducted multi-propeller configurations seems to be small at various dimensionless tip distances ($d/R$) with the height difference fixed (Figure 13b): The $L_{D-MP}$ and $FM_{D-MP}$ show a marginal variation. This indicates that the tip-distance-induced aerodynamic interaction merely has effect in impairing aerodynamic performance within a narrow range of $d/R$ from 0.82 to 0.46 but is negligible over a range of $d/R$ from 1.54 to 0.82.

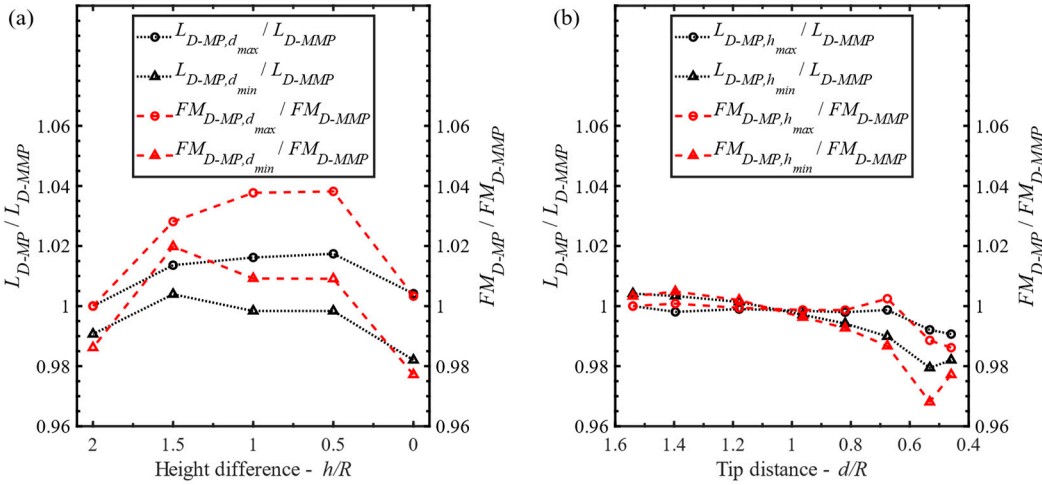

**Figure 13.** (**a**) Lift force and FM efficiency vs. $h/R$ in various ducted multi-propeller configurations: $L_{D-MP, d_{max}}$ and $FM_{D-MP, d_{max}}$ with $d_{max}$ fixed and $L_{D-MP, d_{min}}$ and $FM_{D-MP, d_{min}}$ with $d_{min}$ fixed. (**b**) Lift force and FM efficiency vs. $d/R$ in various ducted multi-propeller configurations: $L_{D-MP, h_{max}}$ and $FM_{D-MP, h_{max}}$ with $h_{max}$ fixed and $L_{D-MP, h_{min}}$ and $FM_{D-MP, h_{min}}$ with $h_{min}$ fixed.

These results are supported by the visualized flow structures of different ducted multi-propeller configurations at cross-sections of $y/R = 1.708$ ($y = 0.205$ m) and $1.183$ ($y = 0.142$ m), the planes in which the centers of propellers P1 and P2 are located as illustrated in Figure 14, as well as the pressure distributions of different ducted multi-propeller configurations on the suction side as shown in Figure 15. The interaction between the upper and lower positioned ducted-propellers at the appropriate height difference is effective and enables one to improve the induced velocity of lower propellers ($h/R = 0.5$ in Figure 14a) as well as the pressure gradient on the duct inlet surface ($h/R = 0.5$ in Figures 14b and 15) with

the tip distance fixed. This is beneficial to improve the lift force production, whereas this interaction weakens at $h/R = 0$ and 2.0 due to the lack of height difference and the too-large distance of the height difference. The interaction among ducted propellers counteracts each other and impairs the downwash-jet because of the interference among them when the tip distance is small with the height difference fixed (Figure 14a), which is harmful to the wake velocity and pressure gradient on propeller and duct surfaces (Figures 14b and 15). Hence, this suppresses the lift force production on the propeller and duct, whereas this interaction is negligible when the tip distance is large because of the downwash-jet separation and interference avoidance owing to the large distance of ducted propellers. In other words, the aerodynamic performance of the ducted multi-propeller can be improved with an appropriate height difference and retained by decreasing the tip distance to a minimal value, which is useful to explore the ducted optimal multi-propeller configuration.

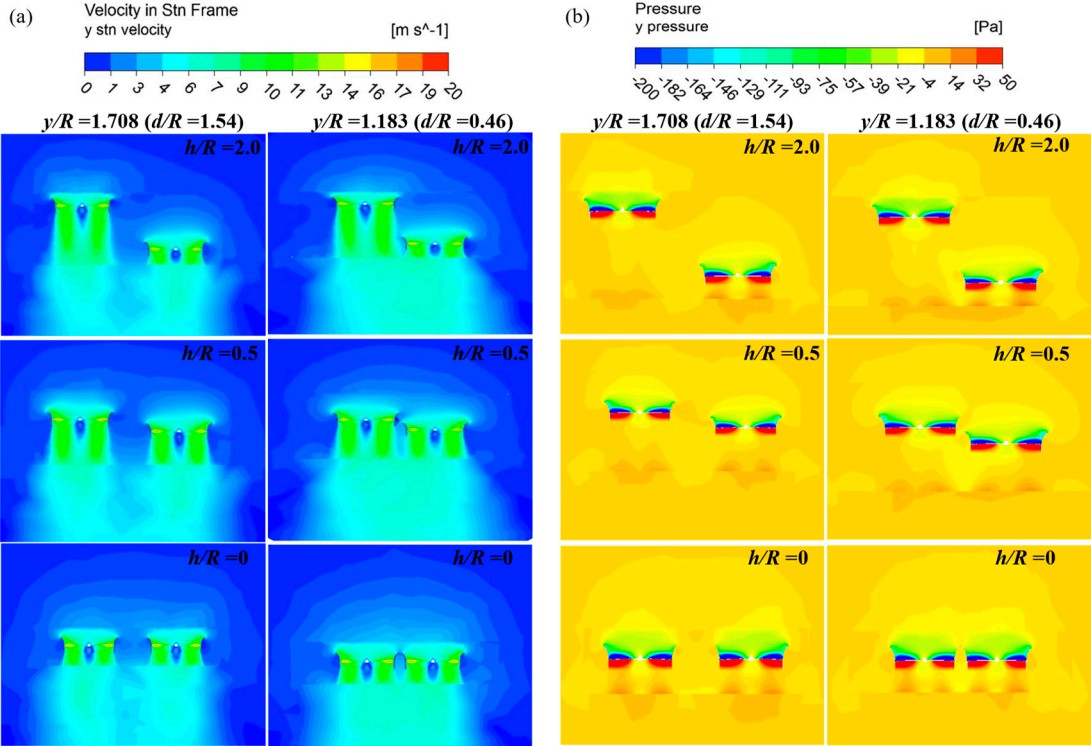

**Figure 14.** (**a**) Iso-speed contours and (**b**) pressure contours in various ducted multi-propeller configurations at cross-sections of $y/R = 1.708$ and 1.183.

### 3.3. Optimization of Ducted Multi-Propeller Configuration

Optimization of the ducted multi-propeller configuration was finally explored based on the $L_{D-MP}$ obtained from various configurations through combining a novel surrogate model with a set of CFD-based simulations. Considering that $FM_{D-MP}$ shares a similar variation trend with $L_{D-MP}$, we thus limited our approach merely to the lift force optimization. As a consequence, the objective function associated with the optimization procedure is defined as:

$$\begin{cases} Max \ L_{D-MP} \\ s.t. \ 0 \leq h/R \leq 2.0, \ 0.46 \leq d/R \leq 1.54 \end{cases} \tag{9}$$

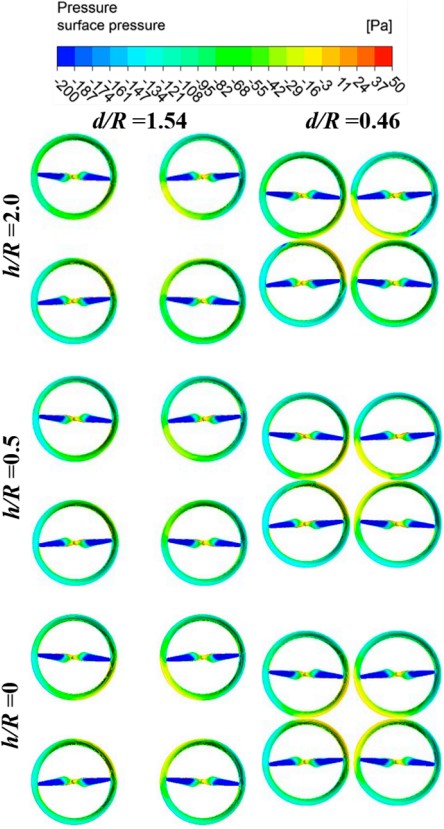

**Figure 15.** Pressure distributions at the suction side in various ducted multi-propeller configurations.

In addition to the CFD-based results of the 22 cases of various multi-propeller configurations conducted in Section 3.2, we further performed an additional 13 cases of CFD simulations to ensure a sufficiently smooth spatial surface of the objective function and hence, an accurate estimation of $L_{D-MP}$, which are summarized in Table 3 and eventually constituted the design space consisting of 35 cases of CFD simulations in total in this process.

**Table 3.** Additional 13 cases of CFD simulations in the surrogate model-based optimization procedure.

| *h*/*R* | *d*/*R* | | | | | | | |
|---|---|---|---|---|---|---|---|---|
| $h/R = 1.5$ | – | – | 1.18 | – | – | 0.68 | – | – |
| $h/R = 1.0$ | – | 1.40 | – | 1.00 | 0.82 | – | 0.53 | – |
| $h/R = 0.6$ | 1.54 | 1.40 | – | – | – | – | 0.53 | 0.46 |
| $h/R = 0.5$ | – | – | 1.18 | – | 0.82 | 0.68 | – | – |

As shown in Figure 16, other than the lift force obtained merely in some discrete points by the CFD simulations because the high fidelity CFD simulation is computationally expensive, the surrogate model method combined with finite CFD-based results is capable of predicting the lift force at each point consecutively and quickly (continuous spatial surface/function) while exploring the optimal lift force accurately and comprehensively in a broad parametric space. The interpolation points and optimal observed points selected in the surrogate modeling are shown in Figure 16a, where the boundary of the observed points is limited to the vicinity of the interpolation points marked with the red dashed frame in the manner of random selection to determine the optimal observed points. The surrogate model-based results utilizing the RBFs model method with the IMQ function are shown in Figure 16b,c, where the blue cross marker "+" denotes the maximum $L_{D-MP}$

of the surrogate model-based results, while the red cross marker "+" represents the maximum $L_{D-MP}$ of the CFD-based results. Figure 16c also exhibits the good fit of the lift force attained from the surrogate model- and CFD-based results, which thus validates the surrogate model-based simulation simultaneously. The comparison among the results as summarized in Table 4 further indicates that an optimal ducted multi-propeller configuration for aerodynamic performance is achieved with $d/R = 0.925$ and $h/R = 0.92$, which is almost identical to the configuration obtained from the CFD-based results with a configuration of $d/R = 1.0$ and $h/R = 1.0$. Moreover, the surrogate model-based result displays that the $L_{D-MP}$ of the ducted optimal multi-propeller configuration can achieve an improvement with an increase rate of 2.1% compared to that of the ducted maximum multi-propeller configuration, and a further increase rate of 17.7% compared to that of the ducted basic multi-propeller configuration. Our results thus point out an optimal and compact ducted multi-propeller configuration design with a minimal tip distance and an appropriate height difference with respect to quadrotor drones, which is capable of markedly improving the aerodynamic performance compared with the ducted maximum multi-propeller configuration.

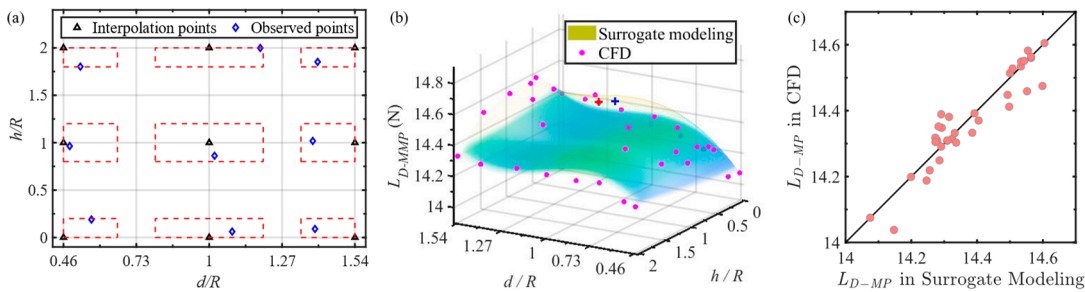

**Figure 16.** Surrogate model-based results: (**a**) Interpolation points and observed points, (**b**,**c**) comparison of CFD-based results and surrogating model-based results.

**Table 4.** Lift forces of the ducted optimal multi-propeller (optimal $L_{D-MP}$) based on CFD simulation and surrogate modeling.

| | Surrogate Modeling | CFD Simulation |
|---|---|---|
| Optimal $L_{D-MP}$ | 14.611 N | 14.605 N |
| Dimensionless value of $d/R$ and $h/R$ at optimal $L_{D-MP}$ | $d/R = 0.925$ ($d = 0.111$ m), $h/R = 0.92$ ($h = 0.110$ m) | $d/R = 1.0$ ($d = 0.120$ m), $h/R = 1.0$ ($h = 0.120$ m) |
| Increase rate of optimal $L_{D-MP}$ compared to $L_{D-MMP}$ | 1.95% | 1.90% |
| Increase rate of optimal $L_{D-MP}$ compared to $L_{BMP}$ | 17.79% | 17.74% |
| Remarks | Optimal $c = 3.3$; $R^2 = 0.9465$, $\bar{e} = 0.0015$, $\sigma_e = 0.0022$. | |

## 4. Conclusions

In this study, we have conducted a systematic analysis of the effect of the ducted multi-propeller configuration on lift force production and FM efficiency while exploring an optimal design of ducted multi-propeller configuration through a combination of CFD-based simulations and a surrogate model. Our main findings are summarized as follows:

1. A high-performance ducted single-propeller design was found, capable of achieving an increase rate of 24.5% in lift force production and 38.1% in FM efficiency compared to the original non-ducted single-propeller model. The ducted multi-propeller configuration model equipped with the high-performance duct design enables a marked improvement in both lift force production and FM efficiency with increase rates of 15.5% and 24.0% in the maximum configuration, 17.7% and 28.0% in the sub-maximum configuration, and even 7.0% and 9.7% in the basic configuration. Our results demon-

　　　strate that ducted propellers can significantly improve both lift force production and FM efficiency of multirotor copters compared to non-ducted multirotor copters.

2. The aerodynamic interaction among ducted multi-propellers shows notable dependency upon two key parameters, the tip distance and height difference between propellers, and thus can be optimized in terms of the ducted multi-propeller configuration. The tip distance has a marginal impact on aerodynamic performances over a range of 0.185 m ($d/R$ = 1.54) to 0.098 m ($d/R$ = 0.82) but impairs the aerodynamic performance within a narrow range ($0.82 \geq d/R \geq 0.46$) with height difference fixed; adjustment of the height difference with tip distance fixed can also improve aerodynamic performance over a certain range of $h/R$ from 1.5 to 0.5.

3. Through combining CFD-based simulations and a surrogate model to determine the effect of the ducted multi-propeller configuration on aerodynamic performance in the quadrotor drone, we found an optimal design of the ducted multi-propeller configuration under the conditions of a minimal tip distance and a specific height difference, which is capable of enabling the maximization of the aerodynamic interaction while reducing the multirotor frame, resulting in an increase rate of about 2% in lift force production and 4% in FM efficiency compared to the original ducted multi-propeller configuration.

In conclusion, inspired by a biomimetic design of multi-rotor configuration in our previous study [19], for the sake of improvement in lift force production and FM efficiency, we demonstrate that some optimal adjustment associated with tip distance and height difference can also benefit the aerodynamic performance of the ducted multi-propeller configuration associated with a multirotor drone. How the current optimal ducted multi-propeller configuration design works in multirotor copters with larger and/or smaller propellers, and how it impacts flight stability and maneuverability, remain unclear and will be our future task.

**Author Contributions:** Y.L., K.Y. and H.L. conceived the concept of this research, conducted a survey of the research plots, and determined the schematic process of the research. Y.L. conducted the simulation and the original draft preparation. H.L. reviewed and conducted the final editing of this manuscript. All authors have read and agreed to the published version of the manuscript.

**Funding:** This work was partly supported by the Grant-in-Aid for Scientific Research of KAKENHI No. 19H02060, 19H00750, JSPS and a Global Prominent Research Program, Chiba University.

**Institutional Review Board Statement:** Not applicable.

**Informed Consent Statement:** Not applicable.

**Data Availability Statement:** All geometric models, simulation information, and surrogate modelling code will be available on request to the corresponding author's email with appropriate justification.

**Acknowledgments:** We are very grateful to the valuable comments and suggestions from Ru Xu, Xuefei Cai and other members of bioinspired flight group in the Graduate School of Engineering, Chiba University.

**Conflicts of Interest:** The authors declare no conflict of interest.

## List of Abbreviations

| | |
|---|---|
| FM | figure of merit |
| UAV | unmanned aerial vehicle |
| CFD | computational fluid dynamics |
| EXP | experiment |
| SP | non-ducted single propeller |
| MP | non-ducted multi-propeller |
| D-SP | ducted single propeller |
| D-MP | ducted multi-propeller |
| D-MMP | ducted maximum multi-propeller |
| D-SMP | ducted sub-maximum multi-propeller |
| D-BMP | ducted basic multi-propeller |
| BMP | non-ducted basic multi-propeller |
| MMP | non-ducted maximum multi-propeller |
| RBFs | radial basis functions |
| IMQ | inverse multiquadric |

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
