# Peer review of "Effect of Ducted Multi-Propeller Configuration on Aerodynamic Performance in Quadrotor Drone"

_drones, doi:10.3390/drones5030101_

Round 1
Reviewer 1 Report
The manuscript entitled “Effect of ducted multi-propeller configuration on aerodynamic performance in quadrotor drone” by Li et al. has been reviewed. It is a nice piece of work that will be of interest to the drone community. In this manuscript, to investigate the aerodynamic performance of the multi-propeller, a ducted multi-propeller design was presented.
- You must proofread extensively for grammar/English.
- No abbreviation in abstract
- The reference should be in numerical order as they appear in the text
- Reference in the abstract is not recommended; please revise (line 16)
- The introduction needs to be revised to reflect the major findings from previous investigations. Explained more about what it means with optimal aerodynamic performance; a detailed discussion/review of past investigations should be included.
- What kind of findings, in what sense? line 167
- Figure 12 (c) needs to be in higher quality
- A better way of presenting Figure 12 would be beneficial. The related discussion on this figure is also unclear; some edit is required to smoothen the reading and clarify the messages embedded in it.
- In what sense? Line 532
- Lack of clarity in Figures 15 and 16
Author Response
Reply to Reviewer 1
We thank this referee for the valuable comments and suggestions. We have largely revised the manuscript accordingly with the revised parts highlighted by red in the revised manuscript. In the following, we give our answers to all the questions in detail.
- You must proofread extensively for grammar/English.
Answer:
The original manuscript has been proofread carefully and extensively, the grammar errors and illogical expressions have been revised in the revised manuscript accordingly.
- No abbreviation in abstract.
Answer:
A list of abbreviations frequently used in the manuscript is added accordingly.
- The reference should be in numerical order as they appear in the text.
Answer:
This is fixed accordingly.
- Reference in the abstract is not recommended; please revise (line 16).
Answer:
This is fixed accordingly.
- The introduction needs to be revised to reflect the major findings from previous investigations. Explained more about what it means with optimal aerodynamic performance; a detailed discussion/review of past investigations should be included.
Answer:
We largely revised the introduction with a specific focus on the issues as suggested by the reviewer. The present study is motivated by our previous study - a proposal of the bioinspired optimal aerodynamic design of non-ducted multi-propeller configuration (Li et al [19]). In this study we aim to further improve the aerodynamic performance of the quadrotor drone by employing a ducted-propeller design while adopting the multi-propeller configuration to optimize the lift force production and FM efficiency. This is explained in detail and extensively in the revised manuscript.
- Li, Y.; Yonezawa, K.; Xu, R.; and Liu, H. A Biomimetic Rotor-Configuration Design for Optimal Aerodynamic Performance in Quadrotor drone. J Bionic Eng. 2021, 18: 824–839.
- What kind of findings, in what sense? line 167.
Answer:
The duct is generally composed of a straight diffuser section and an elliptic or pseudo-elliptic inlet as shown in Figure 3, which is verified to enable a significant improvement in lift/thrust force production and power reduction particularly at low rotational speeds and/or with high disk-loading (Hrishikeshavan et al. [10–12]). We revised the description in the text accordingly.
- Hrishikeshavan, V.; Black, J.; Chopra, I. Design and performance of a quad-shrouded rotor micro air vehicle. J Aircr. 2014, 51(3), 779–791.
- Hrishikeshavan, V.; Black, J.; Chopra, I. Design and testing of a quad shrouded rotor micro air vehicle in hover. AIAA/ASME/ASCE/AHS/ASC Struct Struct Dyn Mater Conf. 2012.
- Hrishikeshavan, V.; Black, J.; Chopra, I. Development of a quad shrouded rotor micro air vehicle and performance evaluation in edgewise flow. Annu Forum Proc - AHS. 2012, 1, 665–684.
- Figure 12 (c) needs to be in higher quality.
Answer:
This is fixed accordingly.
- A better way of presenting Figure 12 would be beneficial. The related discussion on this figure is also unclear; some edit is required to smoothen the reading and clarify the messages embedded in it.
Answer:
We largely revised the text to present Figure 12 in a clarified way, such as:
“These results are supported by the visualized flow fields and pressure distributions (Figure 12) of the non-ducted single propeller model and the high-performance ducted single-propeller model, where the downwash in the ducted-propeller model is weakened rather than that in the non-ducted model (Figure 12a), which results in reducing the propeller-induced lift force, whereas the pressure gradient on the inlet surface augments the lift force production by the duct. Thus, the duct leads to improving the total lift force production in the ducted-propeller model.”
- In what sense? Line 532.
The original sentence in line 532: The maximum multi-propeller configuration in Figure 1b was found capable of achieving the optimal aerodynamic performance.
Answer:
In our previous work (Li et al [19]), we carried out an extensive study about the effect of non-ducted multi-propeller configuration on aerodynamic performance in a quadrotor drone, and found that the maximum multi-propeller configuration in Figure 1b is capable of achieving the optimal aerodynamic performance by a max. increase rate of 9% in lift force compared with a basic non-ducted multi-propeller configuration under hovering state.
- Li, Y.; Yonezawa, K.; Xu, R.; and Liu, H. A Biomimetic Rotor-Configuration Design for Optimal Aerodynamic Performance in Quadrotor drone. J Bionic Eng. 2021, 18: 824–839.
- Lack of clarity in Figures 15 and 16.
Answer:
We largely revised the text to clarify the presentations of Figures 15 and 16 in sections 3.2 and 3.3.
Reviewer 2 Report
- The abstract is too long, the authors may think of reducing the abstract to make it more precise and concise.
- The novelty of the proposed method should be highlighted carefully
- The contribution of the paper is not clear. What is the relevance of this approach compared to other seminal works?
- Please check all Eqs. of the paper and correct them, for example, Eq. 4…
Author Response
Reply to Reviewer 2
We thank this referee for the valuable comments and suggestions. We have largely revised the manuscript accordingly with the revised parts highlighted by red in the revised manuscript. In the following, we give our answers to all the questions in detail.
- The abstract is too long. The authors may think of reducing the abstract to make it more precise and concise.
Answer:
The abstract is largely revised and shortened to present the main findings in a clarified way.
- The novelty of the proposed method should be highlighted carefully.
Answer:
We largely revised the parts relating the novelty of the proposed study in the manuscript. Motivated by a bioinspired optimal aerodynamic design of multi-propeller configuration (Li et al [19]), here we proposed a ducted multi-propeller design, and explore the optimal ducted multi-propeller configuration in terms of lift force production and FM efficiency, through a combination of CFD-based simulations and a surrogate model. An approach to the optimization of the ducted multi-propeller configuration could be performed merely based on CFD-based simulations, which is much computationally expensive. In this study, we proposed a computational framework to resolve the issue. We firstly explored a high-performance duct morphology based on the ducted single-propeller model, and then adopted it in the multi-propeller model. The effect on aerodynamic performance of ducted multi-propeller configuration is examined by CFD-based simulations. We further proposed an integrated approach through combining the CFD-based simulations and a surrogate model, which is verified to be capable of examining the optimal ducted multi-propeller configuration in a broad parametric space in a precise and quick way.
19. Li, Y.; Yonezawa, K.; Xu, R.; and Liu, H. A Biomimetic Rotor-Configuration Design for Optimal Aerodynamic Performance in Quadrotor drone. J Bionic Eng. 2021, 18: 824–839.
- The contribution of the paper is not clear. What is the relevance of this approach compared to other seminal works?
Answer:
We largely revised all the parts relating the new findings of the study throughout the manuscript. Motivated by a bioinspired optimal aerodynamic design of multi-propeller configuration [19], here we proposed a ducted multi-propeller design, and through a combination of CFD-based simulations and a surrogate model, we successfully examined an optimal ducted multi-propeller configuration, which enables a remarked improvement in lift force production and FM efficiency for a quadrotor drone. We believe that this may provide a potential optimal design for multirotor UAVs.
- Please check all Eqs. of the paper and correct them, for example, Eq. 4…
Answer:
This is fixed accordingly.
Reviewer 3 Report
This paper presents a study of ducted propellers on UAVs. Specifically, the authors carried out an integrated simulation-based study by combining CFD-based simulations and a surrogate model to the effect of ducted multi-propeller configuration on aerodynamic performance in quadrotor drones. In my view, the paper is interesting and it may be of interest to the journal readers.
The introduction contains a comprehensive review of the state of the art that largely justifies the contribution, however, I believe that some references should be updated to further improve the paper.
Define all acronyms in first use, even if they are defined in the abstract they should be defined in the main text, e.g. CFD. Check all the manuscript thoroughly. On the other hand, the manuscript contains acronyms that are multiple times defined, e.g. Figure of Merit (FM). Please define it only once.
Figure 2, includes in its caption undefined variables.
Some figures, particularly in the results section must be improved. Vectorized figures would display better.
The paper has language issues, it must be proofread.
Author Response
Reply to Reviewer 3
We thank this referee for the valuable comments and suggestions. We have largely revised the manuscript accordingly with the revised parts highlighted by red in the revised manuscript. In the following, we give our answers to all the questions in detail.
- The introduction contains a comprehensive review of the state of the art that largely justifies the contribution, however, I believe that some references should be updated to further improve the paper.
Answer:
We largely revised the introduction with a specific focus on the issues as suggested by the reviewer. The present study is motivated by our previous study - a proposal of the bioinspired optimal aerodynamic design of non-ducted multi-propeller configuration (Li et al [19]). In this study we aim to further improve the aerodynamic performance of the quadrotor drone by employing a ducted-propeller design while adopting the multi-propeller configuration to optimize the lift force production and FM efficiency. This is explained in detail and extensively in the revised manuscript.
- Li, Y.; Yonezawa, K.; Xu, R.; and Liu, H. A Biomimetic Rotor-Configuration Design for Optimal Aerodynamic Performance in Quadrotor drone. J Bionic Eng. 2021, 18: 824–839.
- Define all acronyms in first use, even if they are defined in the abstract they should be defined in the main text, e.g. CFD. Check all the manuscript thoroughly. On the other hand, the manuscript contains acronyms that are multiple times defined, e.g. Figure of Merit (FM). Please define it only once.
Answer:
A list of abbreviations frequently used in the manuscript is added accordingly.
- Figure 2, includes in its caption undefined variables.
Answer:
This is fixed accordingly.
- Some figures, particularly in the results section must be improved. Vectorized figures would display better.
Answer:
We carefully checked the qualification of all the figures and the corresponding presentations in the text, and we largely revised the related parts accordingly.
- The paper has language issues, it must be proofread
Answer:
The original manuscript has been proofread carefully and extensively, the grammar errors and illogical expressions have been revised in the revised manuscript accordingly.
Round 2
Reviewer 2 Report
I have no more comments. I recommend accepting this article.